# Bisulfite-free epigenomics and genomics of single cells through methylation-sensitive restriction

Christoph Niemöller[1,2,3,4,10], Julius Wehrle [1,2,5,10], Julian Riba[6], Rainer Claus[1,2,7], Nathalie Renz[1,2,7], Janika Rhein[1,2], Sabine Bleul[1,2], Juliane M. Stosch[1,2,4], Justus Duyster[1,2,3], Christoph Plass[8], Pavlo Lutsik[8], Daniel B. Lipka[9], Michael Lübbert[1,2,3] & Heiko Becker [1,2,3✉]

Single-cell multi-omics are powerful means to study cell-to-cell heterogeneity. Here, we present a single-tube, bisulfite-free method for the simultaneous, genome-wide analysis of DNA methylation and genetic variants in single cells: epigenomics and genomics of single cells analyzed by restriction (epi-gSCAR). By applying this method, we obtained DNA methylation measurements of up to 506,063 CpGs and up to 1,244,188 single-nucleotide variants from single acute myeloid leukemia-derived cells. We demonstrate that epi-gSCAR generates accurate and reproducible measurements of DNA methylation and allows to differentiate between cell lines based on the DNA methylation and genetic profiles.

[1] Department of Medicine I, Medical Center - University of Freiburg, Freiburg, Germany. [2] Faculty of Medicine, University of Freiburg, Freiburg, Germany. [3] German Cancer Consortium (DKTK) partner site, Freiburg, Germany. [4] Faculty of Biology, University of Freiburg, Freiburg, Germany. [5] Institute of Digitalization in Medicine, Faculty of Medicine, University of Freiburg, Freiburg, Germany. [6] Laboratory for MEMS Applications, Department of Microsystems Engineering - IMTEK, University of Freiburg, Freiburg, Germany. [7] Department of Hematology and Oncology, Augsburg University Medical Center, Augsburg, Germany. [8] Division of Cancer Epigenomics, German Cancer Research Center (DKFZ), Heidelberg, Germany. [9] Section Translational Cancer Epigenomics, Division of Translational Medical Oncology, German Cancer Research Center (DKFZ) & National Center for Tumor Diseases (NCT) Heidelberg, Heidelberg, Germany. [10] These authors contributed equally: Christoph Niemöller, Julius Wehrle. ✉email: heiko.becker@uniklinik-freiburg.de

Single-cell multi-omics are revolutionizing our understanding of cell-to-cell variability[1–4]. These techniques offer to link genomic, epigenomic, and transcriptomic information from the same cell, and therefore allow to study cell-to-cell variability at unprecedented resolution. However, the simultaneous analysis of genetic variants and DNA methylation in single cells remains challenging. The analysis of single-cell DNA methylation typically relies on methylation-sensitive restriction enzymes (MSRE) or bisulfite conversion. The latter is considered the gold standard for genome-wide methylation analysis, but its application to single cells is hampered by DNA degradation[5], resulting in high dropout levels[4]. Moreover, the bisulfite-induced C > T substitutions impact the ability to concurrently detect gene variants[6]. MSRE-based single-cell approaches typically rely on PCR-based readout and are thus limited in the number of evaluable loci per cell[2].

Here, we report an MSRE-based method with genome-wide readout, which facilitates simultaneous analysis of DNA methylation and genetic variants of the same cell at base-pair resolution (epigenomics and genomics of single cells analyzed by restriction; epi-gSCAR). We devised epi-gSCAR as a multistep single-tube workflow which minimizes DNA loss and reduces the risk of contamination. epi-gSCAR features accurate and reproducible characterization of DNA methylation, while preserving the vast majority of genetic information with moderate incidence of dropouts. Thus, epi-gSCAR allows to identify cell-to-cell differences in the DNA methylation profile, and to assign these differences to a given genotype. The latter is of particular importance for the analyses of leukemias and other cancer specimens, since the malignant cells usually differ from healthy cells and among themselves by the genetic aberrations acquired (i.e., clonal heterogeneity).

## Results

### epi-gSCAR workflow

Analogous to previously published methods[2,7], epi-gSCAR employs digestion using the MSRE HhaI, which results in cleavage of unmethylated recognition sites, while methylated sites stay intact (Fig. 1a). Terminal deoxynucleotidyl transferase (TdT) efficiently adds a 3′ poly(d)A tail to the generated DNA ends, which carry the genome-wide information of unmethylated recognition sites. The resulting tagged restriction enzyme scars serve as priming sites for GAT-oligo(dT)12-adapters[8] containing a constant nucleotide 5′ sequence[9], which are ligated to the free 5′ scar end (Fig. 1a). Subsequently, a second adapter carrying the same constant sequence followed by seven random 3′ nucleotides facilitates quasilinear amplification of the whole genome, including all tagged DNA ends. Thus, epigenetic information represented as intact or scar-tagged HhaI sites (i.e., methylated or unmethylated sites), and the genetic information are conserved and amplified. The resulting primary library amplicons are PCR-amplified, and genetic variants and/or DNA methylation can be analyzed by conventional or next-generation sequencing (NGS).

HhaI is particularly well suited for the application in epi-gSCAR since (i) cleavage generates 3′CG overhangs which are efficiently tailed by TdT; (ii) HhaI is completely blocked by CpG methylation on one (hemi-methylation) or both strands, but not by overlapping methylation (i.e., GCGC)[10,11]; and (iii) the human genome contains 1.69 million HhaI recognition sites, providing superior genome-wide and feature-specific coverage when compared to the Infinium HumanMethylation450 BeadChip (450 K) or MethylationEPIC Kit array (Supplementary Fig. 1). In particular, CpG islands (CGIs) and transcription start sites (TSSs) are strongly enriched for HhaI sites (Fig. 1b). CGI shores, shelves and Fantom5 enhancers show HhaI coverage that is comparable

to the aforementioned conventional cell-bulk assays (Supplementary Fig. 1).

### Application of epi-gSCAR to measure site-specific CpG methylation

First, we applied 27 single cells of the human acute myeloid leukemia (AML) cell line Kasumi-1 to two variants of the epi-gSCAR workflow (Fig. 1a). For all single cells subjected to the epi-gSCAR assay, we could verify successful amplification of library DNA by agarose gel electrophoresis (Supplementary Fig. 2a and Supplementary Data 1). Product quality and fragment size distribution were additionally assessed on a Bioanalyzer (Agilent) High-Sensitivity DNA chip for selected reactions (Supplementary Fig. 2b).

In this first set of cells, we tested whether methylation of individual CpG sites can be assessed by targeted amplification of the loci of interest from the epi-gSCAR library. For this, we utilized step-out PCR, which facilitates isolation of amplicon ends regardless whether the target fragment contains intact HhaI sites or scar-tagged DNA ends[12]. This enables a convenient and cost-effective targeted readout of single-cell DNA methylation by conventional sequencing. We determined the DNA methylation status of two individual CGIs located within the promoters of the long and short isoforms of *DLX4* in single cell K_05 (Fig. 1c). CGI1 was determined to be largely unmethylated, while CGI2 showed strong methylation (Fig. 1d). The results compared well with the DNA methylation levels of six other single cells analyzed by NGS (discussed below), and with Kasumi-1 bulk data derived from 450 K arrays or previously published whole-genome bisulfite sequencing (WGBS)[13] (Fig. 1d, e).

### Application of epi-gSCAR to measure genome-wide CpG methylation

To examine the potential of epi-gSCAR to measure genome-wide DNA methylation, we applied NGS to 7 of the 27 libraries (K_01–K_07). These libraries were sequenced at low depth (10.15–20.78 million mapped reads per cell; 0.41×–0.82× mean depth), achieving up to 18.8% genome coverage at ≥1× depth (Supplementary Data 2). NGS data were then analyzed using a custom bioinformatic pipeline (Supplementary Fig. 3).

To assess the overall quality of the data obtained by epi-gSCAR, we first analyzed the genome-wide methylation profiles of CGIs, gene bodies and five histone marks (Supplementary Fig. 4). The derived profiles were in line with those described in the literature[14,15]. We also compared the single-cell methylation datasets with cell-bulk methylome data obtained from 450 K arrays and from WGBS of Kasumi-1 and observed that the single-cell datasets well resembled the profiles of the cell-bulk controls (Supplementary Fig. 4). HhaI digestion efficiency as assessed by the analysis of non-methylated spike-in DNA was ≥98.3% (Supplementary Fig. 5b–d). Conversely, we confirmed complete digestion blockage of methylated spike-in DNA (Supplementary Fig. 5a).

### Assessment of epigenetic heterogeneity using epi-gSCAR

We next assessed the ability of epi-gSCAR to differentiate between different cell lines based on the DNA methylation profiles. In addition to Kasumi-1, we selected OCI-AML3 as a second cell line which harbors a common, AML typical gene mutation in the *DNMT3a* gene and features a pronounced hypomethylation phenotype[13]. In order to directly compare both cell lines, we applied epi-gSCAR to 80 single cells of each cell line, which resulted in successful amplification of library DNA for all reactions as assessed by agarose gel electrophoresis. Based on the visual verification of single-cell deposition (Supplementary Data 3), we selected 20 cells (K_08–K_27 and O_01–O_20) for NGS analysis using our low-coverage approach (13.67–30.85

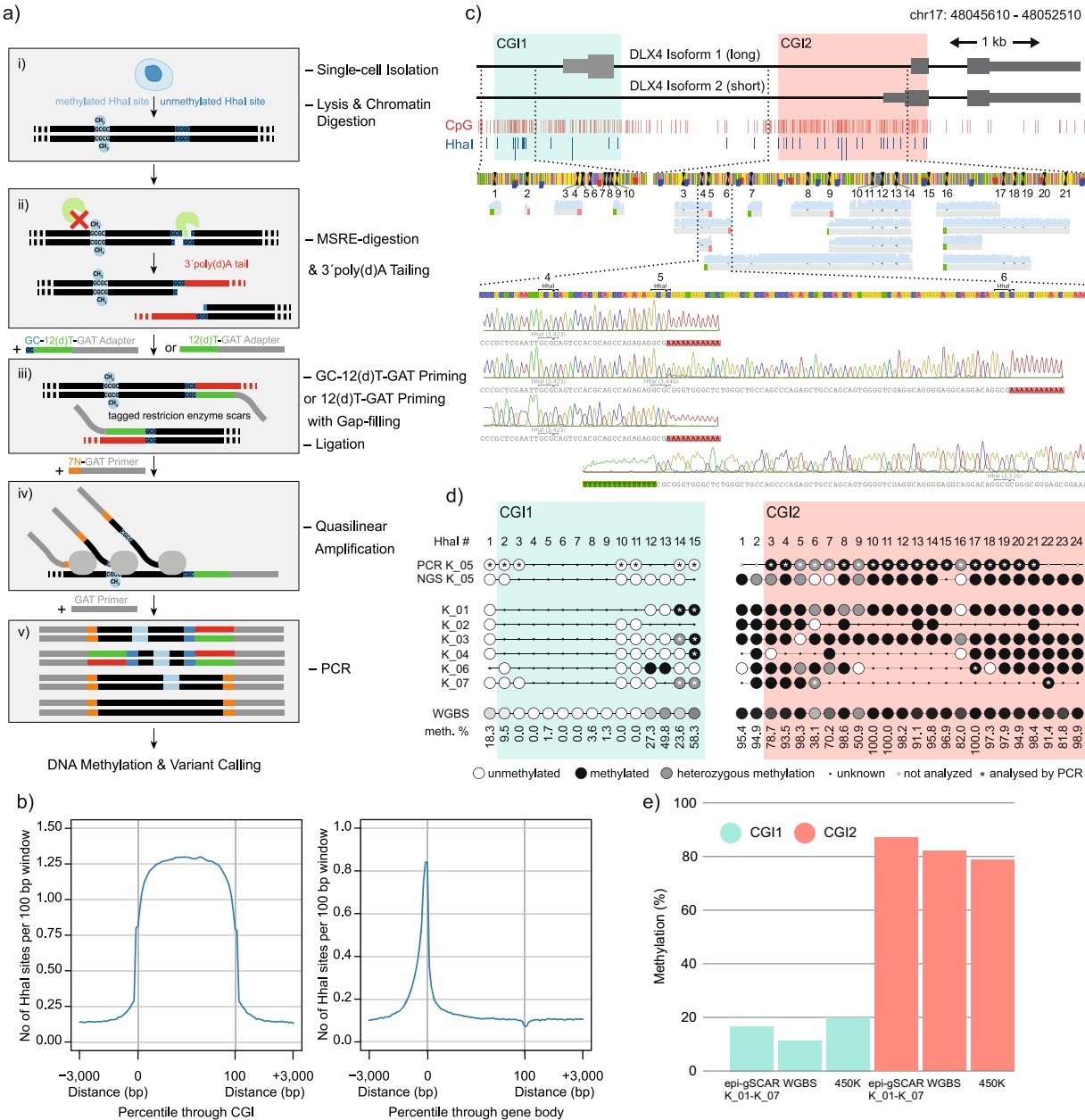

**Fig. 1 epi-gSCAR workflow schematics and methylation readout for the *DLX4* locus. a** epi-gSCAR workflow: single-cell isolation is followed by lysis and chromatin digestion to render DNA accessible for methylation-sensitive restriction enzyme (MSRE) digestion with HhaI (i). Cleavage of methylated HhaI sites (light blue) is blocked, while unmethylated sites (dark blue) are cleaved; the resulting DNA ends are tagged with poly(d)A tails (red) (ii). Poly(d)A tails are primed by anchored (GAT-oligo(dT)12-CG, blue–green–gray, assay variant A) or non-anchored adapters (GAT-oligo(dT)12, green–gray, assay variant B). Anchored adapters were used to limit the length of poly(d)A tails in the library (Supplementary Fig. 8). This is followed by gap filling and ligation, which results in tagged restriction enzyme scars (iii). Random priming by 7N-GAT adapters (orange–gray) facilitates quasilinear amplification of the genome (iv). PCR generates amplicons carrying genetic and epigenetic information (v). **b** HhaI sites in CGIs and around TSSs across 100 bp windows and 3 kb upstream and downstream. **c** Methylation analysis of the *DLX4* locus by step-out PCR followed by Sanger sequencing. *DLX4* locus with CGI1 (green) and CGI2 (red), CpGs (red) and HhaI sites (blue), primer map for analysis of HhaI sites 1–10 (CGI1) and 3–21 (CGI2), and corresponding sequencing reads. Magnification of reads obtained from single cell K_05 (selected for analysis as it demonstrated satisfactory results in initial suppression PCR experiments) corresponding to HhaI sites 4–6 in CGI2 showing intact and tagged-scar HhaI sites: intact HhaI sites are called as having been methylated and poly(d)A-tailed HhaI scar sites unmethylated; presence of both suggests heterozygous methylation. **d** DNA methylation in single cells K_01–K_07 at individual HhaI sites for CGI1 (green) and CGI2 (red) of *DLX4* assessed by PCR and/or NGS (Supplementary Fig. 3), and comparison with Kasumi-1 cell-bulk whole-genome bisulfite sequencing (WGBS) data. Using step-out PCR on single cell K_05, CGI1 was unmethylated at all analyzed HhaI sites (6/6). CGI2 featured high level of heterozygous methylation (14/19 methylated; 5/19 heterozygous methylation). **e** Mean methylation levels of CGI1 (green) and CGI2 (red) for single cells K_01–K_07 (NGS and PCR), Kasumi-1 WGBS and Illumina 450 K array.

million mapped reads per cell; 0.62×–1.37× mean depth; Supplementary Data 2). The generated NGS datasets were used to assess and directly compare DNA methylation and genetic variant features of the cell lines.

For all analyzed cells, we obtained data on 214,634–506,063 CpG dinucleotides (mean: 373,058), corresponding to 0.78–1.85% (mean: 1.36%) of all informative CpG dinucleotides and 13.3–31.6% (mean: 23.2%) of HhaI sites (Supplementary Data 2). For both Kasumi-1 and OCI-AML3, covered CpG dinucleotides provided information on various genomic features, including CGI promoters, non-CGI promoters, orphan CGIs, gene bodies, and intergenic regions, which closely resembled the theoretical distribution of HhaI sites (Supplementary Fig. 6).

Next, we analyzed the sequencing coverage bias and found that the bias was slightly higher than that observed for published MALBAC datasets, while the coverage was more uniform than that of published multiple displacement amplification datasets (Supplementary Fig. 7).

For the second batch of single cells (K_08–K_27 and O_01–O_20), HhaI digestion efficiency was assessed by the analysis of non-methylated random spike-in control DNA (Supplementary Data 2). Reads containing spike-in DNA and covering the unmethylated HhaI control site could be detected in 22 of 40 single-cell libraries. All 22 libraries only contained spike-in amplicons with tailed HhaI scars, which confirmed complete digestion (Supplementary Data 2). Complete digestion for all processed single-cell libraries can be assumed, since incomplete digestion would inevitably result in stochastic concordance decrease (see below).

We next generated and compared single-cell DNA methylation profiles for both cell lines across different histone marks, CGIs, and gene bodies, and could identify clearly distinct methylation profiles for both OCI-AML3 and Kasumi-1, respectively (Fig. 2a, b).

Activating histone marks (H3K9ac, H3K4me3, H3K4me2, and H3K27ac) were associated with low levels of methylation in both cell lines, which is in accordance with previous reports[15]. As expected, the lowest methylation levels were present at H3K4me3 and H3K9ac peaks, which are enriched at active promoters and associated with increased activation of promoter or enhancer regions. For both cell lines, the highest methylation values were measured for Polycomb repression-associated H3K27me3 peaks (Fig. 2a)[14].

Analysis of CGIs revealed the expected depletion of methylation in Kasumi-1 and OCI-AML3 single cells. All analyzed single-cell methylomes exhibited low methylation levels around TSSs and high methylation levels within gene bodies (Fig. 2b).

We then analyzed gene body methylation in correlation with genome-wide gene expression levels by grouping genes based on their genome-wide RNA expression levels in cell bulk into three groups (0–20%, >20–60%, and >60%; Fig. 2b). For both, Kasumi-1 and OCI-AML3 single cells, methylation profiles were in line with the described relationship between gene expression and DNA methylation in gene promoters and gene bodies, i.e., that the depletion of DNA methylation around TSSs and enrichment of methylation in gene bodies correlates with higher expression rates[14]. Indeed, the most highly expressed genes consistently showed the lowest levels of methylation around the TSS and the strongest enrichment of methylation toward the 3′-end of the gene body, although the latter was less prominent when comparing the gene groups of >20–60% and >60% expression in OCI-AML3.

Interestingly, overall DNA methylation levels of profiles for histone marks, CGIs, and gene bodies were apparently lower for OCI-AML3 in comparison to Kasumi-1 single cells.

To compare our single-cell data with cell-bulk methylome data obtained from 450 K arrays and WGBS of Kasumi-1 and OCI-

AML3 cells, we generated synthetic pseudo-bulk samples from the single-cell datasets (Supplementary Fig. 9). Although the number of covered CpGs did not reach saturation, 79.38% ($n = $ 1,277,093) and 74.56% ($n = 1,198,840$) of all informative HhaI sites were covered, using 20 single cells of Kasumi-1 and OCI-AML3, respectively (K_08–K_27 and OC_01–OC_20; Supplementary Fig. 9).

Next, we analyzed how well the pseudo-bulk methylomes (K_08–K_27 and O_01–O_20) resembled the profiles measured in cell-bulk samples. We observed that the pseudo-bulk profiles largely resembled those derived from bulk WGBS and 450 K array data across all genetic features analyzed (Fig. 2a, b). As stated above, discrepancies in the methylation level between the pseudo-bulk and the cell-bulk controls (WGBS and 450 K array) are likely explained by HhaI-based coverage and local sequence context bias (Fig. 1b and Supplementary Fig. 1).

Analysis of single-cell mean methylation values across the entire genome confirmed that OCI-AML3 is strongly hypomethylated when compared with Kasumi-1 (58.8% vs. 79.8%), which is in line with the cell-bulk WGBS and 450 K array data and a previous study[13] (Fig. 3a). To study variation among the single cells, we determined the pairwise CpG concordance across the single-cell libraries, separately for each cell line. Among Kasumi-1 single cells, the concordance was 80.3–93.9% (mean: 87.7%; Fig. 3b), and among OCI-AML3 single cells 77.6–85.2% (mean: 81.0%; Fig. 3c).

To assess global similarities at the CpG level, we compared the synthetic pseudo-bulk methylomes with the 450 K array, and WGBS bulk methylation datasets of Kasumi-1 and OCI-AML3. By calculating Pearson correlation coefficients ($R$), we found that the synthetic bulk methylomes highly correlated with the profiles derived from both cell-bulk assays (Kasumi-1, 450 K $R = 0.95$, WGBS $R = 0.89$; OCI-AML3, 450 K $R = 0.93$, WGBS $R = 0.81$; Fig. 3d). Circos plot representation of genome-wide methylation profiles confirmed a high concordance of the pseudo-bulk datasets and the respective WGBS cell-bulk datasets, and demonstrated remarkably distinct methylation landscapes for Kasumi-1 and OCI-AML3 at the pseudo-bulk and single-cell level (Fig. 3e).

To demonstrate that epi-gSCAR can assign a single cell to its cell line of origin based on the methylation patterns, we first assessed global similarities by calculating pairwise Pearson correlation coefficients across all single-cell datasets. Hierarchical clustering demonstrated that single cells of the respective cell line clustered together (Fig. 3f). We further confirmed this by dimension reduction using UMAP[16] to project the single cells in a two-dimensional space. This again revealed clearly defined Kasumi-1 and OCI-AML3 clusters, and demonstrated that epi-gSCAR can distinguish cells based on their DNA methylation signatures (Fig. 3g).

**Evaluation of genetic heterogeneity using epi-gSCAR.** In order to show that epi-gSCAR can be applied to identify single cells not only based on their individual methylation signature, but also based on genetic variant information we searched for single-nucleotide variants (SNVs) in cells K_08–K_27 and O_01–O_20 using monovar[17]. On average, over 800,000 SNVs (range 498,097–1,244,188) were detected per single cell with ≥10× coverage (Supplementary Data 2). Based on the genome-wide SNV data, we performed UMAP clustering and identified two distinct single-cell clusters corresponding to the two analyzed cell lines (Fig. 3g).

Allelic dropout (ADO) rates were estimated to be as low as 20.7% and comparable to ADO rates achieved by commercially available MALBAC Kits[18,19] (Supplementary Data 2, and

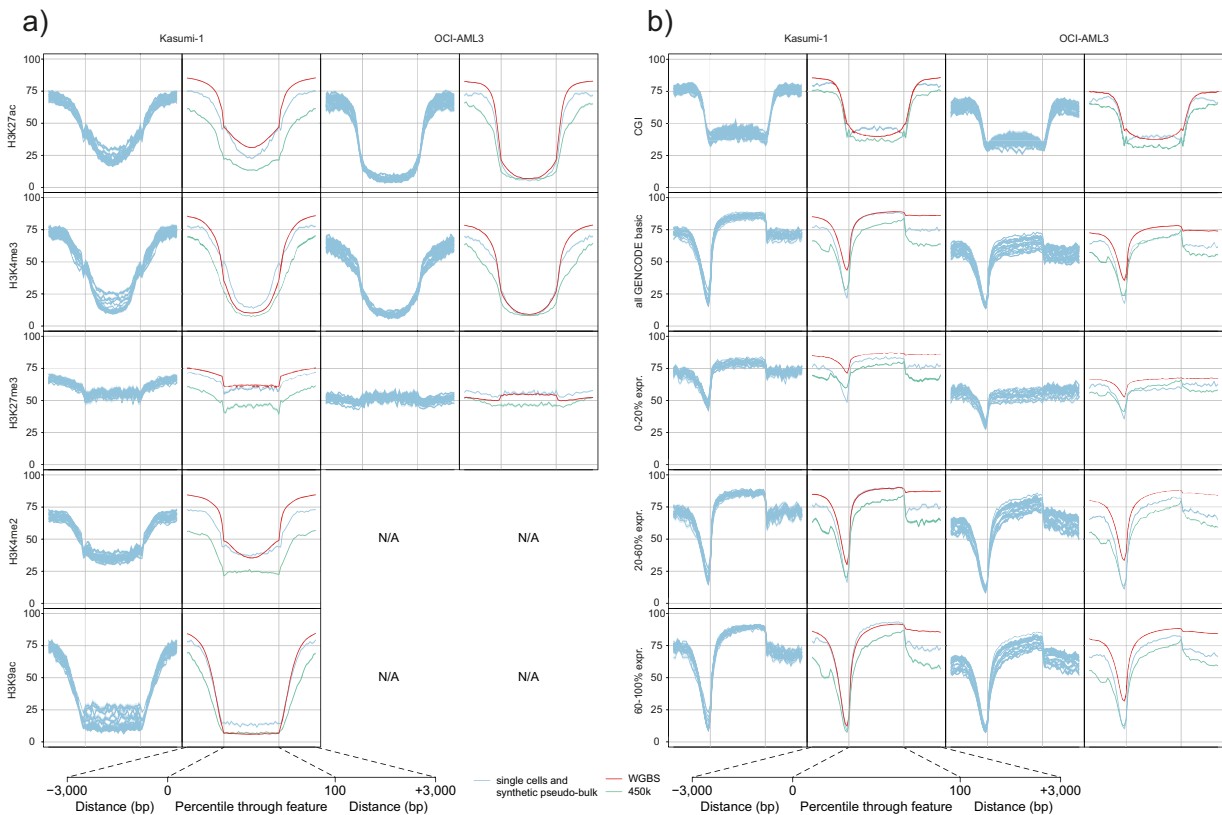

**Fig. 2 Single-cell methylation profiles and comparison of corresponding pseudo-bulk profiles with WGBS and 450 K array cell-bulk profiles.**
**a** Averaged epi-gSCAR methylation profiles (K_08–K_27 and O_01–O_20; light blue lines) and corresponding pseudo-bulk (light blue lines) and cellbulk profiles (WGBS, red lines; 450 K array, green lines) across five histone marks. **b** Averaged epi-gSCAR methylation profiles (K_08–K_27 and O_01–O_20; light blue lines) for CGIs, all genes (all GENCODE basic genes) and genes grouped into three groups (0–20%, >20–60%, and >60–100%) based on their RNA expression level in cell bulk as FPKM (fragments per million mapped reads per kilobase exon), and corresponding pseudo-bulk (light blue lines) and cell-bulk profiles (WGBS, red lines; 450 K array, green lines). For plotting of pseudo-bulk datasets, we used HhaI sites covered in at least 5 of 20 single cells in order to reduce coverage bias. Shown is the mean methylation across 150 bp windows for each feature set and 3 kb upstream and downstream.

Supplementary Figs. 10 and 11). As expected, the ADO rate decreased with increasing coverage, indicating that the current estimates can be reduced by deeper sequencing (Supplementary Fig. 11). Moreover, AML-specific heterozygous or hemizygous point mutations were readily detectable in the epi-gSCAR libraries[20–22] (Supplementary Fig. 12).

## Discussion

We report the development of epi-gSCAR, a bisulfite-free method for genome-wide, base resolution analysis of DNA methylation, and genetic variants at the single-cell level. Applying this method, we achieved consistent site-specific and genome-wide DNA methylation profiling in single cells of two leukemia-derived cell lines. DNA methylation profiles revealed characteristic signatures across various epigenetic elements and methylation patterns correlating with transcriptional gene regulation. We furthermore demonstrate that merged single-cell methylomes highly correlated with cell-bulk data derived from 450 K arrays and WGBS, and that profiles deduced from merged single-cell datasets were highly similar to the cell-bulk control profiles. The overall methylation levels of epi-gSCAR datasets were in accordance with cell-bulk levels and demonstrated that the Kasumi-1 cell line is hypermethylated in comparison with the OCI-AML3 cell lines. Pearson's-based unsupervised clustering and UMAP analysis showed clearly distinct methylation signatures for single cells of the two cell lines, and thus confirmed their epigenetic heterogeneity. Moreover, epi-gSCAR allows for the detection of genetic

variants in addition to DNA methylation in single cells and the discrimination of single cells based on their genetic profile.

Although epi-gSCAR covers a lower number of CpGs compared with single-cell bisulfite sequencing[23], we show that cell-specific epigenetic heterogeneity can be deduced from the epi-gSCAR datasets based on the ~1–2% of CpGs covered. This is in line with the recent finding that measurement of a small stochastically sampled fraction of CpGs (<1%) is sufficient to define the epigenetic state of a single cell[24]. In contrast to bisulfite sequencing protocols, the single-tube epi-gSCAR MSRE-based workflow minimizes nonspecific DNA loss due to bisulfite treatment and sample cleanup, and avoids C > T substitutions, thereby preserving genetic information. In comparison with other published restriction enzyme-based single-cell methods[25,26], which either rely on the combination of methylation-sensitive and -insensitive restriction enzymes or on depletion of unmethylated regions after MSRE digestion and fragment size selection, epi-gSCAR requires only one MSRE to directly retrieve methylation information for both methylated and unmethylated sites at single-nucleotide resolution. This sleek approach to obtain information on the DNA methylation is combined with an efficient WGA technique, and thus allows for NGS-based genome-wide readout of DNA methylation and SNV data.

Further development of epi-gSCAR may include the implementation of a second MSRE in addition to HhaI if, depending on the specific research context, an increase of information density on DNA methylation is necessary. Of high interest will be the integration of epi-gSCAR into current single-cell RNA-

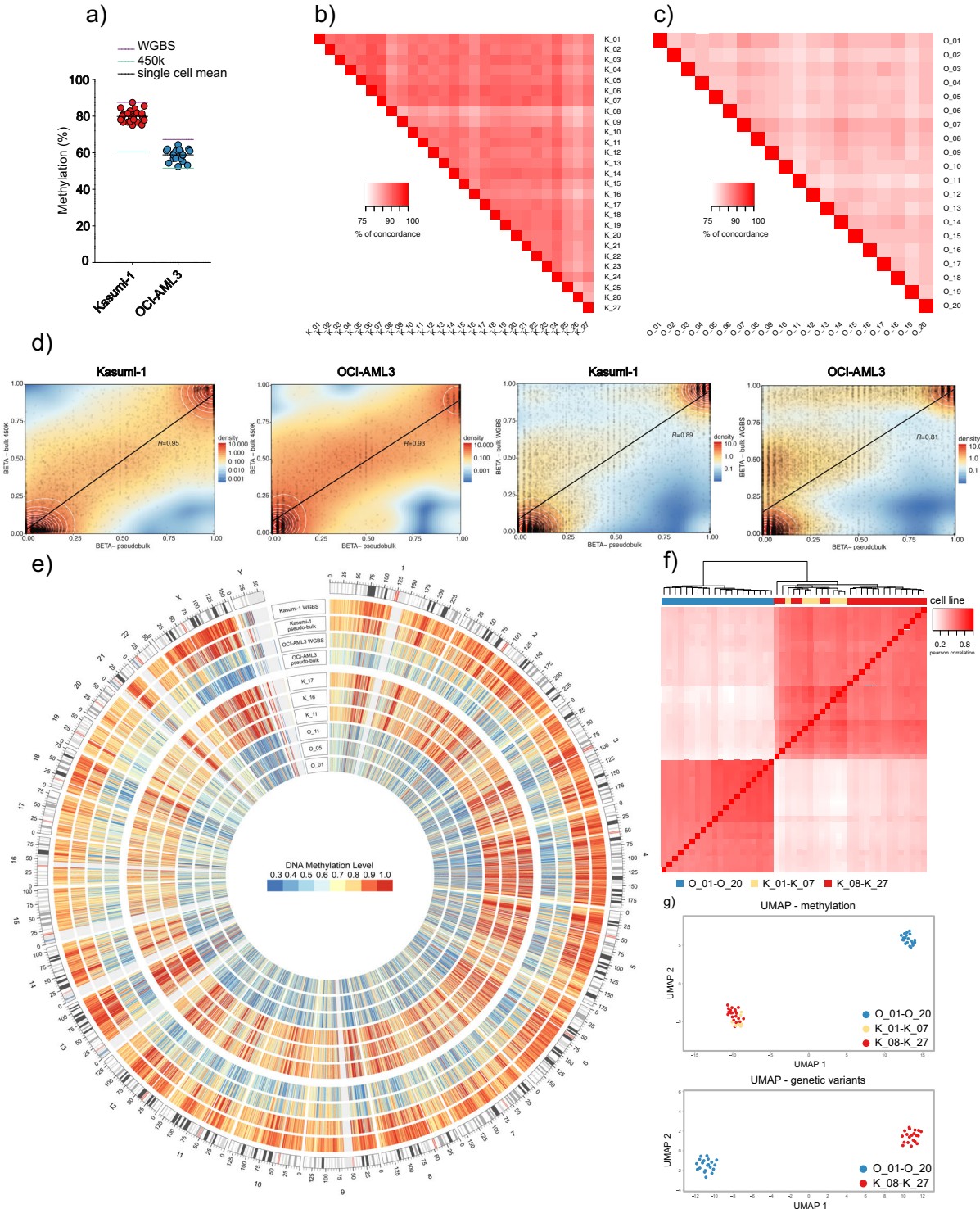

Figure includes panels a) through g) showing DNA methylation analyses. Concordance heatmaps (b, c) with % of concordance scale (75, 90, 100). Density scatter plots (d) for Kasumi-1 and OCI-AML3 comparing BETA-bulk 450K and BETA-bulk WGBS against BETA-pseudobulk (R=0.95, R=0.93, R=0.89, R=0.81). Circos plot (e) showing DNA Methylation Level (0.3 0.4 0.5 0.6 0.7 0.8 0.9 1.0). Pearson correlation heatmap (f) with cell line legend (O_01-O_20, K_01-K_07, K_08-K_27). UMAP plots (g) for methylation and genetic variants.

sequencing protocols, which could be achieved with or without physical separation of mRNA from genomic DNA prior to amplification[1,3,27]. Moreover, the integration of single-cell protein assays is conceivable, which would allow to simultaneously assess a complete picture of the genetic, epigenetic, and functional state of a cell. epi-gSCAR is also readily suitable for combination with a single-cell isolation technique other than the single-cell printer we used, such as fluorescence-activated cell sorting or microfluidic devices.

In conclusion, epi-gSCAR is a multistep bisulfite-free single-tube workflow for the simultaneous and genome-wide characterization of DNA methylation and genetic variants in single cells. epi-gSCAR can readily be applied to assess the phenotypic and genotypic characteristics of single cells, and has the potential to be easily integrated in current and future single-cell assays to further expand the applications of multimodal profiling of single cells, for example, in cancer.

## Methods

**Cell lines**. The AML-derived cell lines Kasumi-1 and OCI-AML3 were provided by the research group of Michael Lübbert (University of Freiburg) who obtained it from DSMZ (Nos. ACC 220 and ACC 582; Braunschweig, Germany). Both cell

**Fig. 3 epi-gSCAR performance and validation. a** Global single-cell methylation levels of Kasumi-1 (red dots) and OCI-AML3 (blue dots), and corresponding mean methylation levels (black lines) in comparison to mean methylation levels of WGBS (violet line) and 450 K array (green line) from cell-bulk samples. **b** Pairwise CpG concordance for all analyzed Kasumi-1 single cells. **c** Pairwise CpG concordance for all analyzed OCI-AML3 single cells. CpG concordance was calculated for all overlapping CpGs between each single-cell pair of each cell line as the fraction of CpGs with the identical methylation state (0, 0.5, or 1). Calculations are based on 66,588–297,161 CpGs for Kasumi-1 single-cell pairs and 148,932–257,837 CpGs for OCI-AML3 single-cell pairs. **d** Correlation between the mean pseudo-bulk methylation and the cell-bulk 450 K array, and WGBS datasets for Kasumi-1 and OCI-AML3. Comparisons consider genome-wide methylation of individual CpGs covered in ≥15 single cells (Kasumi-1 450 K: $n = 6,607$; Kasumi-1 WGBS: $n = 20,000$ of 72,142 CpGs covered by epi-gSCAR in ≥15 single cells; OCI-AML3 450 K: $n = 11,511$; OCI-AML3 WGBS: $n = 20,000$ of 129,153 CpGs covered by epi-gSCAR in ≥15 single cells). **e** Circos plot representation of genome-wide methylation profiles of randomly selected single cells, the pseudo-bulk datasets, and WGBS controls. The heatmaps show average methylation levels for 200 kb windows. Heatmap colors indicate methylation levels from low (blue) to high (red). Tracks from inside to outside represent single cells O_01, O_05, O_11, K_11, K_16, and K_17, OCI-AML3 pseudo-bulk (O_01–O_20), OCI-AML3 cell-bulk WGBS, Kasumi-1 pseudo-bulk (K_08–K_27), and Kasumi-1 cell-bulk WGBS. **f** Hierarchical clustering analysis based on Pearson correlation coefficients for single cells K_01–K_07 (yellow), K_08–K_27 (red), and O_01–O_20 (blue) across 200 kb windows. **g** Multidimensional scaling analysis using UMAP, in which each dot represents a single cell (K_01–K_07, yellow; K_08–K_27, red; and O_01–O_20, blue). Cells are clustered based on the methylation levels across 200 kb windows covered in all single cells (top; $n = 10,555$) or based on genetic variants called at positions covered in all single cells (bottom; $n = 7,027$). For genetic variant clustering, SNV data was converted into a categorical numeric matrix as an input to compute UMAP with the R package ggplot.

lines were cultured in RPMI medium plus 10% fetal bovine serum and 1% penicillin/streptomycin in a humidified 5% $CO_2$ atmosphere.

**Single-cell isolation.** The single-cell printer™ was used as described in detail elsewhere[21,28]. Briefly, prior to single-cell dispensing the cells were washed three times by repetitive centrifugation ($400 \times g$, 2 min) in PBS to yield a final concentration between $10^5$ and $10^6$ cells/ml. For each experiment, a new sterile cartridge with a 40 μm nozzle was loaded with 30 μl sample and mounted on the single-cell printer. The piezo stroke length was set to 10 μm and the downstroke velocity was set to 140 ± 10 μm/s to achieve stable droplets. Individual cells were printed into the wells of a 96-well low-binding PCR plate (FrameStar 96 Well; 4titude) containing 2 μl lysis buffer. For all reactions selected for NGS, we could visually verify the deposition of a single cell (Supplementary Data 3).

Lysis buffer was prepared by premixing of CellsDirect Resuspension Solution with CellsDirect Lysis Enhancer (Invitrogen) at a 10:1 ratio. Electrostatic charges on the plates were neutralized with an ionizing air blower (minION2, SIMCO-ION, The Netherlands). Sample loading and instrument preparation took 5 min on average. Plates were sealed with AlumaSeal II film (Sigma-Aldrich) and frozen at −80 °C and stored for up to 1 week.

**epi-gSCAR.** After thawing, the plates were briefly centrifuged at 1300 r.p.m. and then incubated in a PCR cycler at 75 °C for 10 min with heated lid set to 85 °C. Then, the plate was cooled down on ice and 0.5 μl of Qiagen Protease (2.8 AU/ml) were added to each well. The samples were centrifuged again and incubated for 90 min at 50 °C followed by 30 min at 70 °C (85 °C lid temperature). MSRE digestion and TdT tailing was performed by addition of 3.1 μl reaction mixture containing 0.32 μl HhaI (20 units/μl; NEB), 0.16 μl TdT (20 units/μl; NEB), 0.56 μl 10× CutSmart Buffer (NEB), 0.5 μl 1 mM dATP (Thermo Scientific), and 0.56 μl water. For the first epi-gSCAR experiment (single cells K_01–K_07), we added 0.5 μl of each methylated and unmethylated lambda spike-in control template (60 ag/μl, assay variant A, K_01–K_04; or 6 ag/μl, assay variant B, K_05–K_07; see Supplementary Data 2 for oligonucleotide sequences). For the second epi-gSCAR experiment, we added 1 μl of the non-methylated spike-in control DNA (10 ag/μl, assay variant A, K_08–K_27 and O_01–O_20). After brief centrifugation and gentle vortexing for 1 min, samples were incubated for 120 min at 37 °C followed by 20 min at 75 °C (85 °C lid temperature). For assay variant A, 2.4 μl ligation mixture containing 0.8 μl 10× ThermoPol II Mg-free Reaction Buffer (NEB), 0.66 μl 10 μM GAT-12dt-CG adapter, 0.1 μl Ampligase (5 units/μl; Epicenter), 0.08 μl 50 mM NAD (NEB), and 0.76 μl water were added, and the samples were spun down and mixed. To promote specific annealing and ligation of the anchored adapter, the samples were incubated in a PCR cycler with the following program: 70 °C for 1 min, ramp to 35 °C with 0.1 °C per min, and 35 °C for 10 min. For assay variant B, the reaction mixture contained 0.66 μl of the non-anchored GAT-12dt adapter (10 μM) and additionally 0.05 μl T4 DNA Polymerase Exonuclease Minus (3 units/μl; Lucigen) and 0.33 μl dNTP Solution Mix (10 mM each nt; NEB) in 2.4 μl ligation–gap-filling mixture. To promote specific annealing of the non-anchored adapter, gap-filling and ligation, the samples were incubated in a PCR cycler with the following program: 25 °C for 4 min, 28 °C for 15 min, and 37 °C for 15 min. Then, 10 μl quasilinear amplification reaction mixture was added to the samples containing 1 μl 10× ThermoPol II Mg-free Reaction Buffer (NEB), 0.4 μl dNTP Solution Mix (10 mM each nt; NEB), 0.9 μl 10 μM GAT-7N primer, 0.07 μl 100 mM $MgSO_4$ Solution (NEB), 0.3 μl Deep Vent (exo-) DNA Polymerase (2 units/μl; NEB), 0.01 μl SD Polymerase HS (10 units/μl; BIORON), 3.6 μl Q-Solution (QIAGEN), and 3.72 μl water. The samples were centrifuged and mixed. Plates were incubated for 3 min at 94 °C to denature the DNA and immediately quenched on a 96-well cooling rack (−20 °C). Ten cycles (K_01–K_07) or eight

cycles (K_08–K_27 and O_01–O_20) of quasilinear amplification were performed (25 °C for 2 min, 30 °C for 50 s, 40 °C for 45 s, 50 °C for 45 s, 70 °C for 2 min, 92 °C for 30 s, and 64 °C for 20 s). After quasilinear amplification, samples were further PCR-amplified by addition of 16 μl PCR mix: 1.6 μl 10× ThermoPol II Mg-free Reaction Buffer (NEB), 0.4 μl dNTP Solution Mix (10 mM each nt; NEB), 2.7 μl 10 μM GAT primer, 0.34 μl 100 mM $MgSO_4$ Solution (NEB), 0.2 μl Deep Vent (exo-) DNA Polymerase (2 units/μl; NEB), and 10.76 μl water. The final sample volume of 34 μl was incubated in a PCR cycler with the following program: initial denaturation at 95 °C for 1 min, 19 cycles (K_01–K_07) or 16 cycles (K_08–K_27 and O_01–O_20) of 95 °C for 20 s, 60 °C for 30 s, and 72 °C for 4 min; final extension at 72 °C for 5 min (100 °C lid temperature). All pipetting steps were performed under constant cooling on ice. Positive controls contained 30 pg Kasumi-1 or OCI-AML3 gDNA, which were added to the MSRE digestion and TdT tailing reaction mixture. No template controls contained no single cell. After PCR, product quality was assessed by agarose gel electrophoresis and the samples were cleaned up using the Monarch PCR and DNA Cleanup Kit (NEB), according to the manufacturer's recommendations. Product quality was additionally assessed on a Bioanalyzer (Agilent) High-Sensitivity DNA chip for selected reactions. For the second epi-gSCAR experiment (K_08–K_27 and O_01–O_20), we chose assay variant A for all reactions, since it resulted in the absence of template-independent products in the first experiment (Supplementary Fig. 2).

**NGS library preparation.** epi-gSCAR libraries K_01–K_07 were selected for NGS based on amplification success rate of randomly selected loci across the genome using real-time PCR. A total of 150 ng of purified library DNA were fractionated according to amplicon size into two pools using custom SPRI beads: amplicons with a size range of 400–2000 bp were recovered by incubation of the library DNA with a 15% PEG SPRI bead solution in a ratio of 1:0.75 (6.4% PEG) for 10 min at RT. Small amplicons (range, 100–600 bp) were recovered from the supernatant by addition of a 30% PEG SPRI bead solution to a final PEG concentration of 16% and incubation for 10 min at RT. The large amplicons were then fragmented using 1 μl NEBNext dsDNA Fragmentase in a total volume of 10 μl in 1× Fragmentase buffer for 30 min at 37 °C. The reaction was stopped by addition of 2.5 μl 0.5 M EDTA. Fragmented DNA was then recovered by addition of a 30% PEG SPRI bead solution to a final PEG concentration of 12.5% and incubation for 10 min at RT. SPRI beads of all reactions were washed twice with 150 μl 80% ethanol and eluted with 10 μl NEB Monarch elution buffer. NGS libraries were prepared with the NEBNext Ultra II DNA Library Prep Kit for Illumina (NEB) using different index primers for each single cell (NEBNext Multiplex Oligos for Illumina). Each NGS library was prepared with 10 ng of each pool (small non-fragmented and fragmented amplicons), according to the manufacturer's recommendations with 50% reduction of reaction volumes and tenfold (1:10) adapter dilution. epi-gSCAR libraries K_08–K_27 and O_01–O_20 were randomly selected for NGS. A total of 50 ng of purified library DNA of each single cell and each 50 ng of Kasumi-1 and OCI-AML3 gDNA were sheared in 15 μl 1× TE on a Covaris M220 instrument in a microTUBE, using the 250 bp Target BP program: 80 s, 20% duty factor, 30 W peak incident power, and 50 cycles/burst. NGS libraries were prepared with 50 ng of sheared DNA according to the manufacturer's recommendations with 50% reduction of reaction volumes using different index primers for each single cell (NEBNext Multiplex Oligos for Illumina; Unique Dual Index Primer Pairs). For all NGS libraries, adapters were diluted tenfold (1:10) and fragment size distributions were verified on a Bioanalyzer (Agilent) High-Sensitivity DNA chip to have an average product size of ~400 bp. Libraries were sequenced on an Illumina HiSeq 2000 machine using 125 bp paired-end mode (K_01–K_07) and on an Illumina NovaSeq 6000 System machine, using 150 bp paired-end mode (K_08–K_27 and O_01–O_20).

**Step-out PCR**. Step-out PCR was performed with the HotStarTaq DNA Polymerase (QIAGEN) Kit in 20 µl containing 0.4 µl dNTP Solution Mix (10 mM each nt; NEB), 0.4 µl 1 µM GAT-step-out primer, 0.2 µl 10 µM step-out primer, 0.6 µl 5 µM gene-specific primer, 0.16 µl HotStarTaq DNA Polymerase, and 1 ng epi-gSCAR library template. The reaction was run in a PCR cycler using the following program: initial denaturation at 95 °C for 15 min, 5 cycles of 94 °C for 30 s, 72 °C for 150 s; 5 cycles of 94 °C for 30 s, 70 °C for 150 s; 29–35 cycles of 94 °C for 30 s, 68 °C for 150 s; and final extension at 72 °C for 5 min (100 °C lid temperature). Gene-specific primers were designed using Primer3. Primers were ordered as cartridge or HPLC purified standard oligonucleotides (see Supplementary Data 2 for oligonucleotide sequences).

**NGS bioinformatic analysis**. For the extraction of DNA methylation from epi-gSCAR data, NGS reads were analyzed using a custom bioinformatic pipeline (Supplementary Fig. 3), which was automated using Snakemake (version 5.3.0) in Python (version 3.6). After removal of Illumina adapters, overlapping paired-end reads were merged and nonoverlapping reads were converted to singletons using BBMerge to obtain single-read information (merge rate for epi-gSCAR libraries is between 77 and 85%). Merging is beneficial during scar read identification since each read can be filtered independently of its corresponding pair. This ensures that only reads carrying a tailed scar are being pooled, while preserving most of the paired read information. Next, GAT-Adapter sequences were removed. The resulting preprocessed merged and unmerged reads were filtered for reads containing either 5′ poly(d)T or 3′ poly(d)A-tailed HhaI scars separately (motif: GCGAAAAAA or TTTTTTTCGC; Hamming distance = 1). Poly(d)T and poly(d)A tails were removed, resulting in reads containing 5′ or 3′ HhaI scars, respectively (5′-scar file and 3′-scar file). Separately, poly(d)T and poly(d)A tails were removed from GAT-Adapter-trimmed reads (all-read file). All trimming and filtering steps were performed using BBDuk. Reads were subjected to quality control by FastQC and 5′ or 3′ HhaI scar-containing reads were mapped to the human assembly GRCh37 (hg19) with BWA-MEM separately with soft trimming enabled. Samtools was used to remove secondary and supplementary alignments, and alignments with MAPQ smaller than 10. Alignment intervals were generated with the bamtobed command of bedtools and reduced to the outermost three 5′ or 3′ nucleotides, respectively. Scar intervals were filtered for nucleotide-precise overlap with HhaI sites and CpGs in the human genome, and assigned as cut HhaI sites. In order to identify uncut (intact) HhaI sites, the all-read file was aligned accordingly, since all reads can potentially contain intact HhaI sites. Next, all HhaI sites of the human genome were expanded by 1 bp on either side as a safety margin, and only completely covered intervals were assigned as uncut HhaI sites. Intact sites overlapping with cut HhaI sites (e.g., GCGCG(A)$_n$-3′) were excluded from the output, since complete digestion of sites close to DNA ends cannot be guaranteed. Overlap of uncut with cut sites revealed sites of heterozygous methylation. All other uncut and cut HhaI sites were assigned as methylated or unmethylated, respectively. All CpG or HhaI sites of the human genome, which were covered by WGBS datasets of Kasumi-1 or OCI-AML3, respectively, were defined as informative for epi-gSCAR datasets.

**Preparation of unmethylated and methylated spike-in DNA amplicons**. For K_01–K_07, two spike-in amplicons containing each a single HhaI recognition site were PCR-amplified from lambda DNA and then purified, using the Monarch PCR & DNA Cleanup Kit (NEB), according to the manufacturer's recommendations (see Supplementary Data 2 for oligonucleotide sequences). For use as a methylated control, 1 µg of the spike-in meth lambda PCR product was methylated in 50 µl containing 2 µl M.SssI CpG Methyltransferase (4 units/µl; NEB) in 1× NEBuffer 4, supplemented with 0.3 µl 200× SAM. The reaction was incubated at 37 °C for 16 h, and another 0.3 µl of 200× SAM and 2 µl M.SssI CpG Methyltransferase were added, and the incubation was continued for further 6 h. The amplicon was purified and complete protection from HhaI cleavage was controlled by standard restriction digest protocol with HhaI followed by agarose gel electrophoresis. For K_08–K_27 and O_01–O_20 a random 300 bp spike-in oligonucleotide (eurofins genomics) containing a single HhaI recognition site was amplified and then purified, using the Monarch PCR and DNA Cleanup Kit (NEB), according to the manufacturer's recommendations.

**Evaluation of digestion efficiency of spike-in amplicons**. Methylated and unmethylated exogenous spike-in DNA was used to control for digestion efficiency. Each amplicon contained one HhaI site. For K_01–K_07 we spiked in 30 ag (assay variant A, single cell K_01–K_04) or 3 ag (assay variant B, single cell K_05–K_07) of unmethylated and methylated lambda spike-in amplicon. This corresponds to ~110 or 11 and 60 or 6 molecules of unmethylated and methylated amplicons, respectively. For K_08–K_27 and O_01–O_20, we spiked in 10 ag of randomly unmethylated control DNA corresponding to ~30 molecules (Supplementary Data 2). For calculation of the digestion efficiency, the all-read files were filtered for reads matching corresponding amplicons using BBDuk, and aligned to the enterobacteria phage lambda reference genome or the random control oligonucleotide sequence with BWA-MEM. Then, local coverage of scar or intact reads at the control HhaI site was calculated.

**ChIP-seq analysis**. Quality of raw sequence data was controlled with FastQC and sequencing adapters were trimmed with Trim Galore in automatic detection mode, using standard settings. Preprocessed reads were aligned to the human genome GRCh37 (hg19) using Bowtie2 and peaks were identified with MACS2 callpeak. Peaks were called relative to the respective input controls when available.

**Analysis of MALBAC and MDA datasets**. Publicly available MALBAC and MDA datasets were downloaded from the European Nucleotide Archive (SRS2062840; MALBAC, Yikon, Single Cell YK5) and the Sequence Read Archive (SRR617646; MALBAC, sw480 single cell[9], and SRR5219394; Qiagen Repli-g MDA[29]), mapped with BWA-MEM to the human reference genome GRCh37 (hg19). Both MALBAC datasets were downsampled to read numbers comparable to the epi-gSCAR libraries prior to mapping.

**Kasumi-1 and OCI-AML3 bulk RNA-seq analysis**. Quality of raw sequence data was tested with FastQC and sequencing adapters were trimmed with Cutadapt, using standard settings. Preprocessed reads were aligned to the human genome GRCh37 (hg19) using RNA STAR, and fragments per million mapped reads per kilobase exon were determined using FPKM count of the RSeQC package. Expressed genes were grouped based on their FPKM distribution into three groups (0–20%, 20–60%, and >60% expression).

**DNA methylation analysis using the Illumina 450 K BeadChip array and WGBS**. Illumina 450k Infinium methylation array data was normalized with the beta mixture quantile algorithm and further analyzed using routines from the RnBeads software package[30,31]. For OCI-AML3, publicly available 450 K array data from the Gene Expression Omnibus (GEO) database were used (GSM1670296). WGBS of Kasumi-1 and OCI-AML3 was performed, as described elsewhere[13], the data were kindly provided by David H. Spencer (Washington University School of Medicine, St. Louis, MO, USA).

**SNP array analysis, variant calling, and ADO rate estimation**. For single cells K_01–K_07 mapped all-read epi-gSCAR bam files were used as inputs for variant detection with FreeBayes in simple diploid calling mode. Only mapped reads with mapping quality >30 and base quality score >20 were utilized for variant identification. Minimum coverage was set to ≥10× for single-cell libraries. Calculation of ADO estimates was based on Human SNP Array 6.0 (Affymetrix) data of Kasumi-1 (ref. [21]). SNPs with variant allele frequencies ≥0.45 and ≤0.55 were defined as heterozygous. ADO estimates were calculated as the fraction of SNPs called as heterozygous in SNP array data of Kasumi-1 cell-bulk DNA though called homozygous in single cells for loci with coverage ≥10×. ADO rate estimates were additionally calculated for loci with coverage ≥3× and ≥6× in single cells. For genotyping of single cells K_08–K_27 and O_01–O_20, we processed the all-read epi-gSCAR bam files with monovar using standard settings with consensus-filtering step enabled, which removes variants with support from only one cell. In order to calculate ADO rate estimates, we filtered Kasumi-1 and OCI-AML3 SNP 6.0 datasets for heterozygous SNPs with high confidence values (0.999). For OCI-AML3, publicly available Human SNP Array 6.0 array data was downloaded from the GEO database under accession GSM888549. Genotype calling of SNP 6.0 data was performed using the R package CRLMM. ADO estimates were calculated as the fraction of SNPs called as heterozygous in SNP 6.0 datasets, while called homozygous in single cells for loci with ≥20× coverage.

**Statistics and reproducibility**. For the generation of Fig. 3f, Pearson correlation coefficients (R) were calculated using the tool plotCorrelation.py of the deepTools 3.3 package[32]. For the calculation of Pearson correlation coefficients (R) for Fig. 3d, we used the function stat_cor in package ggpubr R package (v 0.4.0/R version 3.5.2). We applied epi-gSCAR to a total of 214 single cells in four independent experiments and obtained reproducible results among the epi-gSCAR libraries subjected to sequencing (27 single Kasumi-1 single cells and 20 OCI-AML3 single cells).

**Reporting summary**. Further information on research design is available in the Nature Research Reporting Summary linked to this article.

## Data availability

All sequencing data (FASTQ) and processed single-cell DNA methylation data (BEDGRAPH) have been deposited in the Gene Expression Omnibus (GEO) database under accession GSE131723. Publicly available ChIP-seq and RNA-seq datasets used in this study were obtained from the GEO data portal with the following accessions: GSE29225, GSE62847, GSE83660, GSM1844449, GSM1844483, GSM3024903, GSM3032904, GSM3024909, and GSM3032912. MALBAC and MDA datasets were obtained from the European Nucleotide Archive (SRS2062840) and the Sequence Read Archive (SRR617646 and SRR5219394). Publicly available Human SNP Array 6.0 array data was downloaded from the GEO (GSM888549). Kasumi-1 Human SNP Array 6.0

array data is available at EMBL-EBI ArrayExpress (E-MTAB-4950). List of figures that have associated raw data: Figs. 1d, e, 2a, b, and 3a–g, and Supplementary Figs. 4, 6–11.

## Code availability

The bioinformatic pipeline used for readout of methylation information from epi-gSCAR libraries by NGS is illustrated in Supplementary Fig. 3. The code is available from https://github.com/wehrleju/epi-gSCAR.

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

## Acknowledgements

The authors thank David H. Spencer (Washington University School of Medicine, St. Louis, MO) for providing the whole-genome bisulfite sequencing data on Kasumi-1 and OCI-AML3. Moreover, the authors acknowledge the provision of bioinformatic tools and support by the Freiburg Galaxy team (Björn Grüning and Rolf Backofen; Dept. of Computer Science, University of Freiburg). The authors also thank Gabriele Greve (Dept. of Medicine I, Medical Center — University of Freiburg) for the scientific discussions on DNA methylation patterns in cancer. This research was supported by the Translational Research Training in Hematology of the European Hematology Association and American Society of Hematology (H.B.); the German Research Foundation (Deutsche Forschungsgemeinschaft, DFG; SPP 1463 LU 429/8-2 [M.L.]; CRC992 MEDEP C04 [M.L.]; FOR 2674 BE 6461/1-1 A05 [H.B.], LU 429/16-1 A05 [M.L.], LI 2492/3-1 A06 [D.B.L.], and A09 [M.L., C.P.]), the Böhringer Ingelheim Foundation (Exploration Grant [H.B.]), the German Cancer Aid (Deutsche Krebshilfe; 111210 [H.B.]; 110461 [R.C.]; and 70112574 [D.B.L.]), the Berta Ottenstein Fellowship Program of the University of Freiburg (J.W.), the Fördergesellschaft Forschung Tumorbiologie Liquid-Biopsy Initiative (J.W.), the Federal Ministry of Education and Research (Bundesministerium für Bildung und Forschung, BMBF) Eurostars Project E!10257 (Julian Riba), and the German José Carreras Leukemia-Foundation (Deutsche José Carreras Leukämie-Stiftung; R 14/25 [M.L.], 17 R/2019 [H.B.]).

## Author contributions

C.N., J.W., D.B.L., J.D., M.L., and H.B. undertook conception and design of the study, C.N., Julian Riba, N.R., Janika Rhein, S.B., and J.M.S. performed or assisted with molecular and cellular experiments, C.N. and J.W. developed computational pipelines and analyzed the data, P.L. assisted with the computational analyses, R.C., J.D., C.P., P.L., D.B.L., M.L., and H.B. provided conceptual advice, M.L. and H.B. secured funding and supervised the work. C.N., J.W., and H.B. wrote the manuscript. All authors accepted the final version of the manuscript.

## Funding

## Competing interests

Julian Riba is an employee of Cytena GmbH, which produces the single-cell printer used in the study. The remaining authors declare no competing interests.
