## [Peer Review File · Communications Biology]

Reviewers' comments:

Reviewer #1 Remarks to the Author):

This paper reports an approach which permits the identification of single nucleotide variants and methylation in the same single cell using a method the authors call epi-gSCAR (for Epigenomics and Genomics Single Cells Analyzed by Restriction). Their method is based on methylation sensitive restriction enzymes and therefore avoids the high dropout rates and C>T substitutions induced by bisulfite conversion; the choice of HhaI as MSRE permits the unmethylated fragments to retain DNA integrity, while the methylated fragments are not cleaved by HhaI and retain the methylation signal.

1. The authors explored both variants of this approach in 27 single cells from the Kasumi-1 AML cell line. Seven of the libraries generated were taken forward for NGS to measure genome-wide DNA methylation. Were these 7 libraries selected randomly, or, if not, what criteria were applied to make the selection.
 - a. There is the statement that SC5 showed one unmethylated and one strongly methylated promoter for the short and the long isoforms of DLX4, and that these results were consistent with the other six single cells and the bulk data. How was SC5 chosen for presentation?
2. I am still struggling to understand how the single cell data were used to construct the synthetic bulk methylome.
 - a. The top of page 7 suggests that this will be elucidated by Supplementary Figure 10, but that show only the numbers of HhaI sites for "all possible combinations of seven individual epi-gSCAR libraries", considering both the union and the intersection. There would appear to be only a single synthetic bulk methylome used in calculating the correlations with experimental bulk data. What was used?
 - b. The correlations are given as "r", which is fine, but it is only later in the text that the authors reference the Pearson correlation coefficient. This should be referenced the first time it is used.
 - c. The legend for figure 2c suggests that the first reference to correlation on page 7 for the synthetic methylome is for the mean of the 7 SCs over the union of all CpGs covered in the seven single cells, with the overlapping CpGs from the 450K array and from WGBS. The legend for 2d sounds similar, but also contains the pairwise correlations of the individual SCs.
 - i. In 2c, the correlation of the mean of the 7 SCs with 450K and WGBS is presented as 0.95 and 0.91, respectively. In 2d, presenting the hierarchical clustering, the correlations between the means of SC1-7 and 450K and WGBS are 0.93 and 0.92. Why have they changed? What has been defined differently here?
 - ii. The Circos plot in 2f is quite nice. It specifies the use of the overlap CpGs for 450K and WGBS, which matches 2c and presumably matches 2d.
3. The supplementary tables are informative. The supplementary figures require careful attention.
 - a. In Figure S2, the lower two panels for SC2 and SC5 are stated in the legend to be conveying highly similar size distribution profiles. But the figure includes color and a lot of text that is overlapping and unreadable. Can this be cleaned up and still demonstrate the size distribution profiles?
 - b. Figure S3 should show 7 single cells. I have looked carefully. I see the light blue for SC1, but I cannot find the dark blue for SC7, and I can only find curves for six single cells.
 - c. The color legend for Figure S8 shows a light pink coded "N"; I cannot find light pink in the figure.

Reviewer #2 (Remarks to the Author):

The manuscript “Bisulfite-free epigenomics and genomics of single cells through methylation-sensitive restriction” by Niemöller et al describes the combined usage of methylation sensitive DNA cleavage and NGS for DNA methylation analyses on single cell level. The approach is bisulfite-free, which implies less DNA degradation and better reads mapping allowing simultaneous analysis of genetic variants. The restrictive usage of HhaI sites limits the population of analysable CpGs to 1-2%, which could still be enough for some particular studies. The results and the description of methods used, reported in the manuscript, are solid, comprehensive and adequately interpreted. The manuscript is well written and delivers clear messages to the reader.

There are some minor suggestions to the authors:

It is still not clear whether assay variants “A” and “B” deliver similar results or differ in some aspects?

Would be curious to read the author’s speculations about the usability of this method for studying allele-specific DNA methylation particularly at imprinted genomic regions.

What are the potential extensions of the method, which could be implemented further? For example – the combination of HhaI and HpaII enzymes to increase the proportion of informative CpGs.

Recommendation: publish after minor corrections

Dr. Konstantin Lepikhov
University of Saarland
Germany

Reviewer #3 (Remarks to the Author):

In this manuscript, the authors claim that their methodology gives the possibility to perform for the first time the simultaneous DNA-methylation and genetic variant analysis in single-tube and bisulfite-free settings. Indeed, the proposed approach is novel and potentially interesting for the scientific community since it allows single cell analysis of CpGs methylation without bisulfite conversion, therefore reducing the probability of DNA degradation. However, I consider that the total number of cells used for library preparation (7 cells) is not enough to clearly draw any conclusion. I suggest that the authors sequence a higher number of libraries in order to deeper test the reproducibility and the overall performance of the presented method. Additionally, it is known that cell epigenetic state and cell identity can be described by a small stochastically-sampled fraction of CpGs (roughly 1%). Therefore, I suggest the authors to use additional cell lines and demethylation agents (for example Azacytidine) to investigate whether the proposed approach is sensitive enough to allow the recognition of different cell lines and/or epigenetic states.

The authors, without any specific explanation, state that the libraries were sequenced at low coverage. Low-coverage sequencing of many single cells samples is usually the preferred strategy for reconstructing cellular lineages. However, the authors did not investigate here whether combining data from each single cell low-coverage methylome results in an increase genome-wide coverage of CpGs. Probably, 7 single cell methylomes are not enough to answer this question, which can be addressed by sequencing more samples. It would be excellent to compare the combination of several low-coverage single cell methylomes with 1 or few high-coverage single cell methylomes to understand whether pulling data from low-coverage methylomes produced with this methodology is

better than increasing the sequencing coverage.

Finally, as stated in the title, the method should provide analysis of single cell methylation (epigenomics) and genetic variants (genomics). However, the genetic variant option remains quite unexplored and the authors dedicate only a small paragraph to its characterization. As it is now, the genetic variant part does not seem as relevant as to be added in the title, although this extra-possibility that the methodology opens could be potentially interesting. I suggest the authors to explore further this part by, for example, try to identify specific genetic variants characteristic of other cell lines and related to specific pathologies. I would finally suggest to further increase the coverage to verify whether the ADO score additionally decreases and become less variable of what it is actually shown in supplementary Figure 11.

Rebuttal Letter - Reviewer comments and Replies

Author's note: For the ease of comparison a list of the data, which have been added to or moved in the revised manuscript compared with the initial submission, is enclosed after the point-by-point replies.

Referee expertise:

Referee #1: Cancer epigenetics

Referee #2: Single-cell epigenetics

Referee #3: Cancer epigenetics, single cell epigenetics

Reviewers' comments:

Reviewer #1 (Remarks to the Author):

This paper reports an approach which permits the identification of single nucleotide variants and methylation in the same single cell using a method the authors call epi-gSCAR (for Epigenomics and Genomics Single Cells Analyzed by Restriction). Their method is based on methylation sensitive restriction enzymes and therefore avoids the high dropout rates and C>T substitutions induced by bisulfite conversion; the choice of HhaI as MSRE permits the unmethylated fragments to retain DNA integrity, while the methylated fragments are not cleaved by HhaI and retain the methylation signal.

1. The authors explored both variants of this approach in 27 single cells from the Kasumi-1 AML cell line. Seven of the libraries generated were taken forward for NGS to measure genome-wide DNA methylation. Were these 7 libraries selected randomly, or, if not, what criteria were applied to make the selection.

Reply: The initial seven single-cell libraries K_01 – K_07 were selected based on successful PCR amplification of randomly selected loci located across the genome. The single-cell libraries K_08 – K_27 and O_01 – O_20 that have been now added to the revised manuscript were randomly selected after visual verification of single-cell deposition. We have added this information to the manuscript (pages 19 and 20) and hope that this clarifies this issue.

a. There is the statement that SC5 showed one unmethylated and one strongly methylated promoter for the short and the long isoforms of DLX4, and that these results were consistent with the other six single cells and the bulk data. How was SC5 chosen for presentation?

Reply: K_05 was selected as it demonstrated promising results in initial suppression PCR experiments. We have added this information to the legend of Figure 1.

2. I am still struggling to understand how the single cell data were used to construct the synthetic bulk methylome.

a. The top of page 7 suggests that this will be elucidated by Supplementary Figure 10, but that show only the numbers of HhaI sites for “all possible combinations of seven individual epi-gSCAR libraries”, considering both the union and the intersection. There would appear to be only a single synthetic bulk methylome used in calculating the correlations with experimental bulk data. What was used?

Reply: In contrast to the initial submission, in which datasets of seven single Kasumi-1 cells were used to calculate the pseudobulk, the revised manuscript now contains two pseudo-bulk datasets, each of which was generated from datasets of 20 Kasumi-1 or 20 OCI-AML3 single cells, respectively. The Kasumi-1 and OCI-AML3 pseudo-bulk datasets contain the mean beta values for each CpG position covered in at least two single-cell datasets. Accordingly, the pseudobulk datasets are composed of mean values of 2-20 single-cell beta values. We have added this information to the legend of Supplementary Figure 9. Further combinations considering the permutation of up to 20 single-cell datasets are displayed in Supplementary Figure 9. They were calculated in order to demonstrate that the total number of covered HhaI sites increases with the number of analyzed cells and that the number of HhaI sites common to multiple epi-gSCAR libraries decreases when the number of analyzed datasets is increased.

b. The correlations are given as “r”, which is fine, but it is only later in the text that the authors reference the Pearson correlation coefficient. This should be referenced the first time it is used.

Reply: Thank you for the careful reading. The Pearson correlation (R) is now referenced the first time it is used (page 12).

c. The legend for figure 2c suggests that the first reference to correlation on page 7 for the synthetic methylome is for the mean of the 7 SCs over the union of all CpGs covered in the seven single cells, with the overlapping CpGs from the 450K array and from WGBS. The legend for 2d sounds similar, but also contains the pairwise correlations of the individual SCs.
i. In 2c, the correlation of the mean of the 7 SCs with 450K and WGBS is presented as 0.95 and 0.91, respectively. In 2d, presenting the hierarchical clustering, the correlations between the means of SC1-7 and 450K and WGBS are 0.93 and 0.92. Why have they changed? What has been defined differently here?

Reply: For the revised manuscript we updated Figure 2d, which is now Figure 3f. It now only shows the Pearson correlations between all individual single-cell datasets. The Pearson correlations were calculated based on the methylation levels across 200kb windows. The same window size was used for UMAP clustering (Figure 3g). In contrast, the values displayed in Figure 2c, which is now Figure 3d, stem from the calculations for individual CpGs covered in ≥ 15 single cells vs. WGBS and 450k array cell-bulk controls. This difference accounts for the deviations

among the correlation coefficient values. We provide details on the calculation in the Figure Legend.

ii. The Circos plot in 2e is quite nice. It specifies the use of the overlap CpGs for 450K and WGBS, which matches 2c and presumably matches 2d.

Reply: Thank you! Figure 2e, which is now Figure 3e, was updated. In the initial submission, the figure displayed the genome-wide average methylation levels across 200kb windows of seven Kasumi-1 single cells, the pseudobulk as well as the OCI-AML3 and Kasumi-1 WGBS overlap CpG datasets. The figure now shows the genome-wide average methylation levels across 200kb windows in six randomly selected single cells (three of each Kasumi-1 and OCI-AML3), the pseudo-bulk datasets and the WGBS controls.

3. The supplementary tables are informative. The supplementary figures require careful attention.
a. In Figure S2, the lower two panels for SC2 and SC5 are stated in the legend to be conveying highly similar size distribution profiles. But the figure includes color and a lot of text that is overlapping and unreadable. Can this be cleaned up and still demonstrate the size distribution profiles?

Reply: We have adjusted the size of this figure and removed the numbers in order to make it clearly legible.

b. Figure S3 should show 7 single cells. I have looked carefully. I see the light blue for SC1, but I cannot find the dark blue for SC7, and I can only find curves for six single cells.

Reply: We appreciate the careful reading of our manuscript! We agree that the dark blue curve was not well displayed. For the revised manuscript, we have updated Supplementary Figure 3, which is now Supplementary Figure 7. It now shows plots for a different set of single cells with improved profile visibility.

c. The color legend for Figure S8 shows a light pink coded "N"; I cannot find light pink in the figure.

Reply: Again, thank you for the careful reading. In the revised manuscript, we have updated Supplementary Figure 8 and at the same time removed category "N" from the legend, as none of the histograms contained "N" nucleotides.

--

Reviewer #2 (Remarks to the Author):

The manuscript "Bisulfite-free epigenomics and genomics of single cells through methylation-

sensitive restriction” by Niemöller et al describes the combined usage of methylation sensitive DNA cleavage and NGS for DNA methylation analyses on single cell level. The approach is bisulfite-free, which implies less DNA degradation and better reads mapping allowing simultaneous analysis of genetic variants. The restrictive usage of HhaI sites limits the population of analysable CpGs to 1-2%, which could still be enough for some particular studies. The results and the description of methods used, reported in the manuscript, are solid, comprehensive and adequately interpreted. The manuscript is well written and delivers clear messages to the reader.

There are some minor suggestions to the authors:

It is still not clear whether assay variants “A” and “B” deliver similar results or differ in some aspects?

Reply: In our initial experiments, assay variant A resulted in a slightly better coverage of HhaI-rich regions and absence of template-independent products, which was in contrast to assay variant B. Although the observed differences were subtle, we decided to apply assay variant A to the following experiments. Moreover, the hands-on time was less for assay variant A compared with variant B.

Would be curious to read the author’s speculations about the usability of this method for studying allele-specific DNA methylation particularly at imprinted genomic regions. What are the potential extensions of the method, which could be implemented further? For example – the combination of HhaI and HpaII enzymes to increase the proportion of informative CpGs.

Reply: Our attempts to study allele-specific DNA methylation at imprinted genomic regions were unfortunately inconclusive, presumably due to the low sequencing coverage. In order to increase the proportion of informative CpGs we have tested the use of HpyCH4IV (ACGT) in addition to HhaI in epi-gSCAR. Analyses of these datasets are promising and indeed seem to increase the number of informative CpGs. Nevertheless, these results are too preliminary to be included in the current manuscript.

Recommendation: publish after minor corrections

Dr. Konstantin Lepikhov
University of Saarland Germany

--

Reviewer #3 (Remarks to the Author):

In this manuscript, the authors claim that their methodology gives the possibility to perform for the first time the simultaneous DNA-methylation and genetic variant analysis in single-tube and bisulfite-free settings. Indeed, the proposed approach is novel and potentially interesting for the

scientific community since it allows single cell analysis of CpGs methylation without bisulfite conversion, therefore reducing the probability of DNA degradation. However, I consider that the total number of cells used for library preparation (7 cells) is not enough to clearly draw any conclusion. I suggest that the authors sequence a higher number of libraries in order to deeper test the reproducibility and the overall performance of the presented method.

Reply: We very much appreciate the acknowledgment of the potential value of our method. It is an everyday experience that we would like to have a better understanding of the clonal composition of the cancer in our patients, and single-cell multiomics will hopefully pave the way towards this and thereby towards better informed treatment decisions. We would of course appreciate if our assay finds additional application beyond our usual scope of research.

In accordance with the Reviewer's suggestion we have expanded the number of single cells subjected to epi-gSCAR by almost 7x (to a total of 47), and we have included a second leukemia cell line, i.e. OCI-AML3. In detail, the 47 single cells analyzed by epi-gSCAR and considered in the revised manuscript thus comprise the 7 Kasumi-1 cells from the initial submission, and 20 Kasumi-1 cells and 20 OCI-AML3 cells from the revision work.

A list of the extensive data added to the revised manuscript is provided after the replies to the Reviewers. In summary:

(a) we performed comprehensive analyses of DNA methylation profiles across different histone marks and gene bodies for single cells of both cell lines and identified highly distinct profiles across the single cells corresponding to their respective cell line origin.

(b) The additional analyses of the epi-gSCAR DNA methylation datasets confirmed the previously described hypomethylation phenotype of OCI-AML3 and hypermethylation phenotype of Kasumi-1.

(c) Through calculation of Pearson correlation coefficients we corroborated our initial observation that the epi-gSCAR pseudobulk methylome of Kasumi-1 highly correlated with the 450k and WGBS cell-bulk data, and moreover, we could successfully expand this analysis to OCI-AML3.

(d) The ability of epi-gSCAR to differentiate between Kasumi-1 and OCI-AML3 cell lines based on their epigenetic heterogeneity was further verified by hierarchical clustering based on Pearson correlation coefficients and by multidimensional scaling analysis using UMAP.

(e) In order to demonstrate the potential of epi-gSCAR to differentiate between the two cell lines, not only based on the DNA methylation profiles but also based on their genetic profiles, we performed analysis of single-cell genetic variants and confirmed cell line-specific clustering based on the single nucleotide variant profiles.

(f) The genetic variant analyses on the expanded number of single cells also confirmed the promising allelic dropout estimates that were described in the initial manuscript.

In summary, we believe that the aforementioned, additional data now confirm the reproducibility and overall performance of epi-gSCAR.

Additionally, it is known that cell epigenetic state and cell identity can be described by a small stochastically-sampled fraction of CpGs (roughly 1%). Therefore, I suggest the authors to use additional cell lines and demethylation agents (for example Azacytidine) to investigate whether the proposed approach is sensitive enough to allow the recognition of different cell lines and/or epigenetic states.

Reply: Thank you very much for this insightful and encouraging comment. In accordance with the Reviewer's suggestion and as stated above, we have now also subjected the leukemia cell line OCI-AML3 to epi-gSCAR. epi-gSCAR could clearly differentiate between Kasumi-1 and OCI-AML3 cells based on their epigenetic and genetic profiles (Figure 3f and 3g).

Regarding the use of a hypomethylating agent (HMA), we of course agree that the application of epi-gSCAR in the context of HMA treatment is of very high interest. This was one of the main drivers of why we started to develop epi-gSCAR in the first place. Thus, we are more than keen to obtain epi-gSCAR data from in vitro and in vivo HMA-treated leukemic cells. However, for the present revision work we had to decide to not include such data due to the following reasons. First, as the changes induced by HMA treatment are subject of ongoing research themselves, i.e. not well established, we were unsure whether or at what time point of treatment HMA-induced differences are sufficient to really test the capabilities of epi-gSCAR. Second, the number of epi-gSCAR experiments including sequencing (considering replicates and controls as well as the necessity of several time points) would have quickly exceeded our resources. Thus, we sincerely hope that the Reviewer understands our disappointment that we are not able to provide pertinent data, and that the Reviewer can follow our decision that this question was beyond the possible scope of the current manuscript, but rather defines an independent research project.

The authors, without any specific explanation, state that the libraries were sequenced at low coverage. Low-coverage sequencing of many single cells samples is usually the preferred strategy for reconstructing cellular lineages. However, the authors did not investigate here whether combining data from each single cell low-coverage methylome results in an increase genome-wide coverage of CpGs. Probably, 7 single cell methylomes are not enough to answer this question, which can be addressed by sequencing more samples. It would be excellent to compare the combination of several low-coverage single cell methylomes with 1 or few high-coverage single cell methylomes to understand whether pulling data from low-coverage methylomes produced with this methodology is better than increasing the sequencing coverage.

Reply: We initially decided to sequence the libraries at a low coverage as we expected that this will provide sound data at costs that are affordable to us and others who want to apply the method. In line with the Reviewer's recommendation, we also aimed for the sequencing of a few libraries with a higher coverage. However, we decided to do this after the low coverage sequencing analyses of the increased number of single cells are completed. Unfortunately, resources (also impacted by the coronavirus pandemic) and consequential time constraints precluded the planned high coverage sequencing. However, and as assumed by the Reviewer, we indeed observed that the higher the number of single-cell methylation datasets included in the analyses, the higher is the genome-wide coverage of CpGs (Supplementary Figure 9). Moreover, based on 20 single cells we achieve a coverage of 79.38% (Kasumi-1) and 74.56% (OCI-AML3) of all informative HhaI sites, respectively, which is slightly higher than observed in the initial submission, where we achieved 68.89% coverage when the pseudo-bulk was calculated from 7 Kasumi-1 single cell datasets.

Finally, as stated in the title, the method should provide analysis of single cell methylation (epigenomics) and genetic variants (genomics). However, the genetic variant option remains quite unexplored and the authors dedicate only a small paragraph to its characterization. As it is now, the genetic variant part does not seem as relevant as to be added in the title, although this extra-

possibility that the methodology opens could be potentially interesting. I suggest the authors to explore further this part by, for example, try to identify specific genetic variants characteristic of other cell lines and related to specific pathologies. I would finally suggest to further increase the coverage to verify whether the ADO score additionally decreases and become less variable of what it is actually shown in supplementary Figure 11.

Reply: We are grateful for the Reviewer's appreciation of the possible, additional read-out of genetic data. Indeed, the possibility for the combined read-out of DNA methylation and genetic data from the same single cell is of utmost relevance in cancer considering the high genetic heterogeneity among clones in a single cancer, especially among older patients and/or after several lines of treatment, an area of research we focus on clinically. epi-gSCAR will allow to assign a distinct DNA methylation profile to a genetically defined clone. We added a pertinent statement to the introduction on pages 4 and 5.

We also expanded the genetic data through the analyses of the single cells that were additionally subjected to epi-gSCAR. In line with the Reviewer's recommendation, we had described the detection of KIT and TP53 mutations as exemplary aberrations characteristic for Kasumi-1 cells in the initially submitted manuscript. In the revised manuscript, we have now added the detection of a DNMT3A Mutation as an aberration specific to OCI-AML3 (supplementary Figure 12).

In addition to the possibility to differentiate between the cell lines based on their DNA methylation profiles (Figure 3g, upper panel) and in line with the Reviewer's suggestion, we also included a new set of analyses of the single-cell genetic variants that demonstrates the possibility to identify single cells of each analyzed cell line based on their genetic profile (Figure 3g, lower panel).

As described above, we were unfortunately not able to sequence single-cell libraries at increased depth in order to verify the observations illustrated in the original Supplementary Figure 11. However, computationally, we compared all single cells with each other for the variants detected via epi-gSCAR in order to further characterize the ADO rate estimates (calculated as the fraction of SNPs called heterozygous in bulk DNA but hemi/homozygous in single cells for loci with coverage $\geq 20x$). In order to use a similar number of Kasumi-1 and OCI-AML3 cells we restricted these analyses to the additionally analyzed 40 cells (Supplementary Figure 10). Both data sets are shown in Supplementary Figures 10 and 11, and they confirm the moderate ADO rate observed with epi-gSCAR.

Data added to or moved in the revised manuscript compared with the initial submission

New data added to the revised manuscript

Note: A total of 40 Kasumi-1 and OCI-AML3 single cells (K_08 – K_27 and O_01 – O_20) were subjected to epi-gSCAR and subsequent analyses, in addition to the 7 Kasumi-1 single cells (K_01 – K_07) considered for the initial submission.

- Averaged epi-gSCAR methylation profiles of Kasumi-1 and OCI-AML3 single cells (K_08 – K_27 and O_01 – O_20) and comparison of the pseudo-bulk profiles of each cell line with WGBS and 450K array data from cell bulk samples stratified for histone mark, CGI and gene expression. (Figure 2a and 2b)
- Comparison of mean global methylation levels of Kasumi-1 and OCI-AML3 single cells analyzed with epi-gSCAR and methylation levels of WGBS and 450k array data from cell bulk samples. (Figure 3a)
- Pairwise CpG concordance analyses for all analyzed Kasumi-1 single cells (K_01 – K_27). (Figure 3b)
→ These data replace: Pairwise CpG concordance analyses for Kasumi-1 single cell K_01 – K_07. (initial submission: Supplementary Figure 7)
- Pairwise CpG concordance analyses for all OCI-AML3 single cells. (Figure 3c)
- Correlation between the mean pseudo-bulk methylation and the cell bulk 450K array and WGBS datasets for Kasumi-1 and OCI-AML3. (Figure 3d)
→ These data replace: Correlation of mean single-cell methylation with 450K and WGBS bulk datasets. (initial submission: Figure 2c)
- Circos plot representation of genome-wide methylation profiles of randomly selected Kasumi-1 and OCI-AML3 single cells (among K_08 – K_27 and O_01 – O_20), the pseudo-bulk datasets and WGBS controls. (Figure 3e)
→ These data replace: Circos plot representation of genome-wide methylation profiles and distribution of SNPs in Kasumi-1 single cells K_01 – K_07. (initial submission Figure 2e)
- Hierarchical clustering analysis based on Pearson correlation coefficients for all analyzed single cells (K_01 – K_27 and O_01 – O_20) across 200kb-windows. (Figure 3f)
→ These data replace: Hierarchical clustering analysis based on Pearson correlation coefficients for single cell K_01 – K_07. (initial submission: Figure 2d)
- Multidimensional scaling analysis using UMAP considering methylation levels across 200kb-windows or genetic variants called at positions covered in all Kasumi-1 and OCI-AML3 single cells. (Figure 3g)
- Oligonucleotide sequences, sequencing statistics, number of covered CpGs, number of detected SNVs and unmethylated control detection for K_08 – K_27 and O_01 – O_20 single cells analyzed by epi-gSCAR. (Supplementary Table 1)
→ These data extend: Sequencing statistics for single cell K_01 – K_07 single cells analyzed by epi-gSCAR and for cell-bulk sequencing of non-amplified control Kasumi-1 gDNA (initial submission: Supplementary Table 1)
- Number of SNPs called heterozygous in single cells K_08 – K_27 and O_01 – O_20 with ≥ 20 coverage and thereof estimated allelic dropout rates (Supplementary Table 1).
→ These data extend: Allelic dropout rate estimates for single cell K_01 – K_07 (initial submission: Supplementary Table 2)

- Visualization of single-cell deposition for all single cells cells (K_01 – K_27 and O_01 – O_20) subsequently analyzed by epi-gSCAR and NGS. (Supplementary File 1)
→ These data replace: Visual verification of single-cell deposition for single cell K_01 – K_07. (initial submission: Supplementary Figure 4)
- Distribution of covered CpGs from all single cells (K_01 – K_27 and O_01 – O_20) analyzed by epi-gSCAR across six genomic features. (Supplementary Figure 6)
→ These data replace: Distribution of covered CpGs from seven single cells (K_01 – K_07) analyzed by epi-gSCAR across six genomic features. (initial submission: Supplementary Figure 6; now: Supplementary Figure 6)
- Lorenz plot depicting amplification biases for four single-cell epi-gSCAR datasets, two MALBAC datasets, one multiple displacement amplification (MDA) control and two non-amplified gDNA controls. (Supplementary Figure 7)
→ These data replace: Lorenz plot depicting amplification biases for seven single-cell epi-gSCAR datasets, two MALBAC controls and two non-amplified epi-gSCAR controls. (initial submission: Supplementary Figure 3)
- Number of CpGs for union and intersect of all possible combinations of individual epi-gSCAR datasets (2 to 20 single cells) for Kasumi-1 and OCI-AML3 (K_08 – K_27 and O_01 – O_20). (Supplementary Figure 9)
→ These data replace: Number of HhaI sites for union (red) and intersect (blue) of all possible combinations of seven individual epi-gSCAR libraries. (initial submission: Supplementary Figure 10)
- Visualization of the allelic dropout rate estimates in the epi-gSCAR datasets for Kasumi-1 and OCI-AML3 single cells which are stated in Supplementary Table 1. (Supplementary Figure 10)

Data moved within the manuscript without new data having been added

- Averaged methylation profiles for single cell K_01 – K_07 and of WGBS and 450k array data from cell bulk samples (initial submission: Figure 2a and b; now: Supplementary Figure 4)

REVIEWERS' COMMENTS:

Reviewer #2 (Remarks to the Author):

The authors are to be commended for undertaking a large amount of additional work to address the previous review. They have addressed my previous concerns, and in performing the extensive amount of additional work suggested by Reviewer 3, they have substantially strengthened the paper. I have no further concerns.

Reviewer #3 (Remarks to the Author):

The revised version of the manuscript “Bisulfite-free epigenomics and genomics of single cells through methylation-sensitive restriction” by Niemöller et al now contains numerous improvements and additional data (more SCs analysed, additional cell line, deeper genome data analyses, etc.). The reviewers comments were adequately considered and answered. The manuscript can be published without additional revision.